# A Newly Validated HPLC-DAD Method for the Determination of Ricinoleic Acid (RA) in PLGA Nanocapsules

**DOI:** 10.3390/ph17091220

**Published:** 2024-09-17

**Authors:** Lucas Rannier M. de Andrade, Larissa F. dos Santos, Débora S. Pires, Érika P. Machado, Marco Antonio U. Martines, Maria Ligia R. Macedo, Teófilo Fernando M. Cardoso, Patrícia Severino, Eliana B. Souto, Najla M. Kassab

**Affiliations:** 1Laboratory of Pharmaceutical Technology, Faculty of Pharmaceutical Sciences, Food and Nutrition, Federal University of Mato Grosso do Sul, Campo Grande 79070-900, MS, Brazil; larissa_fsanto@yahoo.com.br (L.F.d.S.); debora.s@ufms.br (D.S.P.); erika.pontes@ufms.br (É.P.M.); teofilo.cardoso@ufms.br (T.F.M.C.); 2Institute of Chemistry, Federal University of Mato Grosso do Sul, Campo Grande 79070-900, MS, Brazil; marco.martines@ufms.br; 3Pharmaceutical Sciences, Food and Nutrition College, University of Mato Grosso do Sul, Campo Grande 79070-900, MS, Brazil; ligia.macedo@ufms.br; 4Institute of Technology and Research (ITP), Tiradentes University, Ave. Murilo Dantas, Farolândia, Aracaju 49032-490, SE, Brazil; patricia_severino@itp.org.br; 5UCD School of Chemical and Bioprocess Engineering, University College Dublin, Belfield, D04 V1W8 Dublin, Ireland; eliana.souto@ucd.ie

**Keywords:** castor oil, ricinoleic acid (RA), high-performance liquid chromatography, PLGA nanocapsules, validation

## Abstract

The assessment of ricinoleic acid (RA) incorporated into polymeric nanoparticles is a challenge that has not yet been explored. This bioactive compound, the main component of castor oil, has attracted attention in the pharmaceutical field for its valuable anti-inflammatory, antifungal, and antimicrobial properties. This work aims to develop a new and simple analytical method using high-performance liquid chromatography with diode-array detection (HPLC-DAD) for the identification and quantification of ricinoleic acid, with potential applicability in several other complex systems. The method was validated through analytical parameters, such as linearity, limit of detection and quantification, accuracy, precision, selectivity, and robustness. The physicochemical properties of the nanocapsules were characterized by dynamic light scattering (DLS) to determine their hydrodynamic mean diameter, polydispersity index (PDI), and zeta potential (ZP), via transmission electron microscopy (TEM) and quantifying the encapsulation efficiency. The proposed analytical method utilized a mobile phase consisting of a 65:35 ratio of acetonitrile to water, acidified with 1.5% phosphoric acid. It successfully depicted a symmetric peak of ricinoleic acid (retention time of 7.5 min) for both the standard and the RA present in the polymeric nanoparticles, enabling the quantification of the drug loaded into the nanocapsules. The nanocapsules containing ricinoleic acid (RA) exhibited an approximate size ranging from 309 nm to 441 nm, a PDI lower than 0.2, ζ values of approximately −30 mV, and high encapsulation efficiency (~99%). Overall, the developed HPLC-DAD procedure provides adequate confidence for the identification and quantification of ricinoleic acid in PLGA nanocapsules and other complex matrices.

## 1. Introduction

*Ricinus communis* L., commonly referred to as castor bean (mamona in Brazil), is a plant widely grown in South America, Africa, and India, because of the regions’ optimal temperatures for cultivation. This promotes germination and increases the growth of the plant [1]. The main constituent of castor oil is ricinoleic acid (RA) (12-hydroxyoctadec-9-enoic acid) (RA), approximately 89–92% (Figure 1) [2,3,4]. Brazil, China, and India are the world’s leading producers of castor oil, primarily exporting it to the United States and the European Union [5,6].

According to CONAB/2019, 47.9 thousand hectares of castor beans were cultivated in Brazil in the 2018/2019 harvest, mainly in the states of Bahia and Mato Grosso [7]. Market research by Data Bridge Market Research, in 2023, described global revenue of around USD 2.19 billion from castor oil production. This amount is expected to increase considerably, reaching approximately USD 3.68 billion by 2031, with a compound annual growth rate (CAGR) of 6%. This advance is driven by the growing demand in various sectors, such as cosmetics, biofuels, and pharmaceuticals [8].

Ricinoleic acid has been used in several industries, mainly in the agricultural, cosmetic, food, and pharmaceutical sectors [9,10,11]. Several studies have shown promising antimicrobial, anti-inflammatory, antioxidant, and gastric-protective activities, making RS an interesting product for the pharmaceutical industry [5,11,12,13]. In addition to ricinoleic acid, other compounds, such as carotenoids, phospholipids, tocopherols, or phytopherols, may also be present in castor oil, so that the purification process by enzymatic hydrolysis, chemical treatment, or transesterification leads to obtaining RS with high degrees of purity for use in the various industrial sectors mentioned [14].

The Food and Drug Administration (FDA) approved castor oil as a laxative medication. Castor oil is a GRAS (Generally Recognized as Safe) product used as a food additive, flavouring substance, and/or adjuvant. Despite its reported therapeutic benefits, castor oil has limitations to its use, due to potential toxicological risks. Excessive doses can cause side effects, such as nausea, vomiting, and cramps. A case reported by Steingrub et al. (1998) [15] describes a pregnant woman who experienced amniotic fluid embolism and cardiorespiratory arrest, associated with castor oil ingestion. This incident, along with another report of a pregnant woman experiencing labor induction, cardiorespiratory arrest, and coma after ingesting approximately 400 mg/kg of castor oil, highlights the potential dangers of its excessive use [14]. The toxic effects were attributed to ricin, a glycoprotein present in the oil that can inactivate ribosomes in mammals. Despite this, the level of toxicity of this glycoprotein is influenced by the route of exposure. Inhalation is the most dangerous route, with a fatal dose (LD_50_) as low as 3–5 µg/kg. In contrast, the oral LD_50_ is significantly higher (20 mg/kg). Additionally, being a macromolecule, ricin has limited absorption through the skin [16,17].

The use of nanotechnology is a promising tool for the encapsulation of bioactives to protect against thermal degradation or oxidation and, at the same time, increase their bioavailability through different delivery methods. Some polymeric nanocapsules are particularly effective for this purpose. They are both biodegradable and biocompatible, and they facilitate efficient drug delivery to specific cells and tissues. This approach can increase the effectiveness and stability of RA while also reducing systemic toxicity [18,19,20]. The choice of polymer for the development of nanocapsules may be linked to their applicability, in addition to factors, such as biocompatibility and biodegradability. These polymers may be from natural sources (e.g., albumin, chitosan, alginate and gelatin [21,22,23,24]) or obtained synthetically (e.g., polylactides (PLA), polycaprolactone (PCL), poly (malic acid) (PMLA), poly (methacrylic acid) (PMAA) or poly (lactide co-glycolides) (PLGA) [25,26,27,28]).

According to the International Council for Harmonization (ICH) guidelines on the Requirements for Pharmaceutical Products for Human Use, new analytical methods must be properly validated for detecting and quantifying impurities in new medicines. Validation involves assessing parameters, such as linearity, specificity/selectivity, limit of detection (LOD), limit of quantification (LOQ), precision, accuracy, and robustness. Various methods can be employed for validation, including High-Performance Liquid Chromatography (HPLC), gas chromatography (GC), high-performance thin-layer chromatography (HPTLC), capillary electrophoresis (CE), or supercritical fluid chromatography (SFC) [29,30]. HPLC stands out as one of the most popular techniques for validation, due to its ability to optimize the elution of mobile phases for efficient component separation. This optimization, coupled with the control of parameters such as the pH, temperature, and mobile phase composition, enables HPLC to achieve well-defined resolutions and appropriate analysis times [31]. The Diode Array Detector (DAD) is widely utilized in HPLC analysis for its ability to simultaneously record peaks across a chosen wavelength range. Additionally, the DAD offers temperature control, ensuring stable baseline measurements and cost-effectiveness compared to chromatographs coupled with mass spectrometers [32,33].

HPLC is a popular technique for identifying and analyzing drugs, including those incorporated into nanostructured systems [33,34,35]. Its advantages include the ability to detect very small quantities (low detection limits), cost-effectiveness, and rapid analysis times [28,29,30]. Plante et al. (2009) [36] conducted a study analyzing the profile of castor oil using HPLC-CAD (Charged Aerosol Detection), particularly useful for detecting compounds with weak UV absorbance or complex mixtures. However, the study did not specifically address the validation of the characterization technique and instead focused on the castor oil profile. Therefore, this study focuses on the optimization and validation of an HPLC-DAD (Diode Array Detector) method specifically for the detection and quantification of RA (a crucial bioactive compound) in polymeric nanocapsules [28,33].

This research aims to develop a customized chromatographic method that not only increases the sensitivity and accuracy of AR detection but also facilitates the direct assessment of encapsulation efficiency, an essential qualitative and quantitative parameter in nanocarrier development. By advancing the use of AR in a stable nanostructured system through this validated methodology, this study contributes significantly to both the field of analytical chemistry and the practical application of nanotechnology in drug delivery systems.

## 2. Results and Discussion

### 2.1. Chromatography

Before proceeding with the validation of the method, specific chromatographic parameters, i.e., peak symmetry, theoretical plate numbers, and retention time, were evaluated to generate a simplified and time-saving method that would also cut down the analysis costs. The initial chromatographic analyses by HPLC-DAD were carried out using RA standard in isocratic mode. For this purpose, various concentrations of the mobile phase consisting of acetonitrile and water (pH 3) were tested to obtain the best separation of the substance under analysis. After testing various proportions and flow rates, the best results were achieved with acetonitrile in a 65:35 *v*/*v* ratio, eluted at a flow rate of 0.8 mL/min, with a column temperature of 25 °C, an injection volume of 20 µL, and detection wavelengths of 200, 210, 220, and 240 nm. The maximum absorption peak of ricinoleic acid occurs at 237 nm (Appendix A). Under these set conditions, the peak recorded for RA was detected at ~7.6 min (Figure 2). In addition to retention time, the other evaluated parameters were the symmetry factor (S) and the theoretical plate number (N). The S factor measures the symmetry of a peak to its axis. In the chromatographic analysis of RA, an S value of 1.22 was found. According to the European Pharmacopoeia [37] for perfectly symmetrical peaks, the recommended value should be S < 2; our results are then within the acceptable limit. The theoretical plate number (N) found in this chromatographic analysis was 10.949. This parameter reflects the efficiency of the column in sharpening the peaks, which means that the greater the number of plates, the better the separation of the analyte [38].

### 2.2. Method Validation

#### 2.2.1. Linearity

Linearity was analyzed using a calibration curve to assess the ability of the analytical method to provide results that directly correspond to the concentration of the analyte being analyzed, by observing the existence of a linear relationship between the area values and the concentration of the samples. To assess the linearity of the curves in triplicate, seven concentrations ranging from 3.125 to 95 µg/mL were prepared. The linearity of the method was evaluated through linear regression, assessment of the Y-axis intercept, slope of the regression line, and determination of the correlation coefficient (r). The method under study exhibited a well-defined calibration curve (Figure 3), from which the regression equation of the line was derived (y = 0.7838x + 1.0206) with a correlation coefficient (r) of 0.9991, in agreement with the values specified by the ICH (r > 0.999) [39]. Thus, the method demonstrated linearity within the range of the tested concentrations, allowing the analysis of RA content in the nanocapsules.

#### 2.2.2. Limits of Quantification (LOQ) and Detection (LOD)

The detection and quantification limits were determined based on the signal-to-noise ratios of three concentrations of RA standard (0.391 µg/mL; 0.781 µg/mL; and 1.5 µg/mL) below the lowest point of the calibration curve (3.125 µg/mL). In this study, the LOQ was found to be 3.37 µg/mL, and the LOD obtained was 1.112 µg/mL. These LOQ and LOD values were lower than those reported by Hassan and Hetta (2019) [40], who developed a high-performance thin-layer chromatographic (HPTLC) method for the quantification of RA in castor oil. Despite the differences in methodologies, the significance of achieving low LOD and LOQ values relies on the ability to detect and quantify smaller amounts of the analyte present in nanostructured systems.

#### 2.2.3. Specificity

The specificity (Figure 4) was evaluated by comparing the chromatograms of samples of blank nanoparticles (without RA) and RA standard solutions and polymeric nanocapsules containing RA (NCRA2). To quantify the amount of RA in the nanocapsule formulations (NCRA2), a 200 µL aliquot of the sample was mixed with 200 µL of acetonitrile and subjected to sonication for 10 min (solution 1). A volume of 200 µL of the mobile phase (acetonitrile/water 65:35 *v*/*v*) was added to solution 1, filtered, and analyzed by HPLC. The result peaks corresponding to RA had retention times between 7.4 min and 7.6 min, for both RA standard solutions and for polymeric nanocapsules containing RA. Therefore, the method exhibited specificity, as no interfering peaks were observed near the RA peaks.

#### 2.2.4. Precision

Precision translates the measure of the degree of repeatability of an analytical method under normal operating conditions, and the results are expressed as the percent operation standard deviation (RSD). The results are shown in Table 1.

The results obtained for precision (repeatability), as shown in Table 1, were 1.93% for intra-assay precision and 2.43% for inter-assay precision. These results indicate that our analytical method is precise, since AOAC (2016) [41] recommends an acceptable RSD limit of 7.3% for concentration ranges of 100 ppm (mg/kg) and 5.3% for higher levels.

#### 2.2.5. Robustness

Robustness allows one to identify small and deliberate changes that may or may not affect the method. To analyze the robustness, samples of RA standard at a concentration of 25 µg/mL were used. The variations in flow rate, mobile phase concentration, and pH were studied, and the obtained values are represented in Table 2.

According to the results of the robustness assessment, changing the flow rate to 1.0 mL/min and 0.6 mL/min resulted in changes in the retention time, peak area, and peak symmetry. Although the flow rate adjustment to 0.6 mL/min produced a larger peak area and consequently higher content, it also led to an increase in peak symmetry compared to the normal conditions used for method validation. In contrast to the increase in flow to 1.0 mL/min, which led to a decrease in area, the change in the mobile phase resulted in different values of peak area, concentration, and symmetry for gradient concentrations of 60:40 (*v*/*v*) and 70:30 (*v*/*v*). By varying the pH (4) of the gradient, it was also possible to record a significant change in the concentration and peak area; however, stability in the symmetry of the peaks compared to normal conditions could be maintained, remaining at 1.71. According to the US Pharmacopoeia (1994), peak symmetry values between 0.8 and 1.8 are acceptable for quantification [42]. The obtained results show that the method was not robust with the proposed modifications for the analytical parameters and that the term “small and deliberate modifications” in the analytical parameters must indeed be respected, to avoid making the method impracticable for RA analysis.

#### 2.2.6. Accuracy

Accuracy was determined by calculating the average percentage recoveries for the analyte in three varying concentrations and the relative standard deviation, providing insight into the difference between obtained and accepted result deviation [43]. The accuracy analysis was carried out by adding the commercial RA to three solutions of standard RA at different concentrations (7, 20, and 30 µg/mL). The results presented in Table 3 show an average recovery of 100.04%, 96.92%, and 103.77% with a relative standard deviation (RSD) of 3.14%. The results are in line with those recommended by AOAC (2016) [41], which relates to an accuracy range between 90 and 107% for 100 ppm analyte concentrations and a range of 80–110% for 10 ppm analyte concentrations. Therefore, it was possible to conclude that the results obtained confirm the accuracy of the method.

### 2.3. Gas Chromatography of Ricinoleic Acid (Commercial)

To evaluate the purity of commercial RA used in the preparation of polymeric nanocapsules, a gas chromatography analysis was carried out. The chromatogram of RA is presented in Figure 5. The chromatogram shows the largest peak (around 26 min) relating to RA, representing a purity of 79.95%. Other represented peaks refer to fatty acids and other methyl esters, also identified in the sample. According to Rajalakshmi et al. (2019) [44], RA is the main component of castor oil, representing up to 90%; however, factors, such as the region of cultivation, seed size, type of soil and extraction, and purification method, can vary the RA content.

### 2.4. Particle Size, Polydispersity Index and Zeta Potential of Nanocapsules Containing Ricinoleic Acid (RA)

For the process of developing nanocapsules containing RA in different concentrations, analysis was carried out to determine the average size, polydispersity index, and zeta potential (ZP). DLS (dynamic light scattering) and ZP analyses are fundamental techniques for the evaluation of nanostructures because their measurements demonstrate the efficiency of the development method and the stability of the developed nanocapsules [45]. The results of the average hydrodynamic size of the developed nanocapsules are represented in Figure 6 and Appendix A.

The PLGA nanocapsules containing RA at different concentrations, i.e., Blank NP, NCRA1, NCRA2, NCRA3, and NCRA4, showed an average hydrodynamic diameter of 246 ± 2, 309 ± 3, 359 ± 4, 391 ± 2, and 441 ± 2 nm, respectively. The obtained results were expected for the used production procedure, which typically leads to nanocapsules sizes between 80 and 900 nm [46].

Another evaluated parameter was the polydispersity index (PDI) of polymeric nanocapsules containing RA (Figure 6), with values ranging from 0.158 ± 0.022 to 0.28 ± 0.027. The PDI measures the distribution of particle sizes within the formulation, where values close to 0.0 indicate perfect uniformity and values approaching 1.0 indicate high polydispersity. According to the literature, PDI values below 0.2 are generally acceptable for nanoparticles [47]. The results for our RA-loaded nanocapsules align with these expectations, demonstrating excellent uniformity and promising formulation stability.

Figure 7 and Appendix A show the zeta potential values obtained for the nanocapsules produced. Zeta potential (ZP) is an important measurement that is evaluated through the theoretical calculation of the electrostatic potentiometer of the hydrodynamic shear surface and not by measuring the surface charge of a particle. ZP is also very important for the development of nanocapsules, being an indication of the stability of the colloidal system undergoing aggregation or deposition. ZP is one of the main parameters that translates the colloidal stability of the systems, which can influence the in vivo distribution and cytotoxicity [48,49]. Zeta potential values above ± 30 mV demonstrate that nanocapsules tend not to aggregate due to high electrostatic repulsive forces [50]. The ZP values of blank NP, NCRA1, NCRA2, NCRA3, and NCRA4 were −29 ± 10.5, −30 ± 1.05, −25 ± 0.6, −23± 1.2, and −22 ± 0.35, −23.4 ± 1.2 mV, respectively. Previous studies reported similar values of ZP for PLGA/PVA nanocapsules, attributing these negative values to the presence of polyvinylpyrrolidone (PVA) in the formulation [51,52]. In conclusion, our nanocapsules encapsulating RA presented satisfactory sizes, PDI and ZP, suggesting their physical–chemical stability with a lower risk of aggregation. This result is also interesting for future biological tests, since the stability of nanocapsules may change the release profile of loaded RA [53].

The encapsulation efficiency (EE%) of polymeric PLGA nanocapsules containing RA was evaluated by HPLC analysis to confirm the encapsulation of the active into the polymeric matrix. The EE% results are presented in Table 4**.** The nanocapsules showed high encapsulation values (greater than 99%). This high encapsulation efficiency may be attributed to the low solubility of RA and its high logP (6.19) [54]. The high encapsulation efficiency translates the success of the method chosen for the development of nanocapsules containing RA. A study conducted by Froiio et al. (2019) [55] developed and evaluated antimicrobial PLGA nanocapsules containing essential oils from *Citrus bergamia* (bergamot) and *Citrus sinensis* (sweet orange). The authors reported high encapsulation efficiency values between 28% and 84% for nanocapsules containing bergamot oil, and 48% to 96% for nanocapsules of PLGA containing citrus oil, achieving a 90% bacterial prevention rate for the developed nanocapsules.

### 2.5. Nanoparticle Morphology Characterization

After development, it was assessed that all produced nanocapsules had satisfactory size, PDI, zeta potential values, and encapsulation efficiency. However, for transmission electron microscopy (TEM) analysis, the NCRA1 sample was selected because it contained the least amount of polymer and RA, had a lower PDI value than the other samples, and a zeta potential value closer to −30 mV.

The obtained TEM images (Figure 8) show nanocapsules with a spherical shape and excellent size distribution, corroborating the data obtained for the PDI, which showed the monodispersity of the systems. Furthermore, the encapsulation of RA into nanocapsules can also be confirmed by their well-defined polymeric walls and core (Figure 8a,b). To analyse the distribution, approximately 100 particles were considered for plotting the histogram (Figure 8c). The histogram was adjusted with the log-normal distribution curve (red line), which presented an average size of 191 ± 54 nm. The nanoparticle sizes obtained by TEM are similar to those achieved by DLS. The molecular mass of the polymer, surfactant and/or the preparation method are some of the main parameters that can change nanoparticle sizes. Furthermore, the addition of the active ingredient influences the size of the nanospheres. As a rule, hydrophobic active ingredients lead to the formation of smaller nanocapsules compared to hydrophilic active ingredients. Therefore, the interaction between the solvent, the active ingredient, and the polymer is important for nanoparticle size variation and drug encapsulation [56].

## 3. Materials and Methods

### 3.1. Materials

Commercial ricinoleic acid (purity ~80%) was kindly donated by the company Azevedo Óleos (São Paulo, SP, Brazil), while the standard ricinoleic acid (298.5 g/mol; purity ≥ 98%) was purchased from (Cayman Chemical, Ann Arbor, MI, USA). Poly(lactic-co-glycolic acid) (PLGA, Mw 40,000–75,000), sorbitan monostearate (Span 60^®^), and HPLC-grade acetonitrile were purchased from Sigma-Aldrich (St. Louis, MO, USA). Polyvinyl alcohol (PVA, 89.5% hydrolyzed) was purchased from Êxodo Científica (Sumaré, São Paulo, Brazil). The water used was purified in a Milli-Q Plus system (Millipore, Bedford, MA, USA).

### 3.2. Chromatographic Conditions

The identification and determination of RA encapsulated within PLGA nanocapsules was confirmed using a high-performance liquid chromatography (HPLC) system equipped with a Diode Array Detector (DAD) (Dionex Ultimate 3000; Thermo Fisher Scientific, Waltham, MA, USA) connected to a microcomputer with Chromeleon^®^ 7.1 software—Chromatography Data System. Chromatographic separation occurred in isocratic mode at a flow rate of 0.8 mL/min a C18 Advantage ARMOR™ column (150 × 4.6 mm and 5 µm particle size, Analytical Sales and Service, Pompton Plains, NJ, USA). The mobile phase was composed of acetonitrile/water (65:35 *v*/*v*) with the pH adjusted to 3.0, using 1% phosphoric acid. Detection was performed at 200, 210, 220, and 240 nm. The injection volume was 20 µL, and the run time was 10 min. Analyses were carried out at room temperature (25.0 ± 1.0 °C).

### 3.3. Method Validation

The analytical method was validated according to the standards set by the ICH guidelines [57]. The proposed validation parameters were linearity, selectivity, precision, limit of quantification (LOQ), and limit of detection (LOD).

#### 3.3.1. Linearity

Linearity was estimated by constructing analytical curves. Solutions were prepared from the RA standard at 250 µg/mL [57,58]. Aliquots of these solutions were precisely measured and transferred to 10.0 mL volumetric flasks, and the final volumes were brought to the mark with the same acetonitrile mixture. Then, solutions in a concentration range of 3.125 to 94 µg/mL were obtained. The results were subjected to linear regression analysis, from which analytical curves, straight line equations, and correlation coefficients for RA standard were determined.

#### 3.3.2. Limit of Quantification (LOQ) and Detection (LOD)

The LOD is the lowest concentration of the analyte that can be detected using the analytical method but not necessarily quantified. The LOQ is the lowest concentration of the analyte that can be measured with acceptable precision and accuracy under experimental conditions. The determination of LOD and LOQ was based on the relationship between the standard deviation (SD) of the response, which comprised the SD of the y-intercept of the regression lines and the angular coefficient of the calibration curves (*a*). Equations (1) and (2) mathematically describe this relationship for the respective limits:(1)$$LOD=3.3\times \frac{SD}{a}$$
(2)$$LOQ=10\times \frac{SD}{a}$$
where “*SD*” is the estimate of the standard deviation of the response and “*a*” is the slope or the angular coefficient of the analytical curve.

#### 3.3.3. Selectivity

Selectivity was evaluated by comparing the chromatograms of samples obtained from the supernatant of blank PLGA nanocapsules (without RA), RA standard solutions, and samples containing RA. Solutions containing standard RA, commercial RA and RA obtained from the developed nanocapsules were prepared to determine any possible interferences from the excipients present in the developed formulation.

#### 3.3.4. Precision

Precision was assessed by evaluating both repeatability (intra-day) and intermediate precision (inter-day). Repeatability was confirmed by analyzing six determinations of the same solution prepared from a standard sample at a concentration of 25 µg/mL, all on the same day by the same analyst under consistent chromatographic conditions. Intermediate precision was evaluated by two different analysts performing sextuplicate analyses on two separate days, using the same concentration. Results were reported as standard deviation (SD) and relative standard deviation (RSD). Accuracy was assessed by calculating the RSD values, as described using Equation (3).
(3)$$RSD=\left(\frac{SD}{Mean}\right)\times 100$$

The intra-assay (repeatability) and inter-assay (intermediate precision) should not exceed 5%. The analyses were determined according to the “Validation of Analytical Procedures” guidelines (ICH, 1995) [39].

#### 3.3.5. Robustness

The robustness of the method was determined by making variations in the parameters. The three selected variables were the flow (1.0 mL/min and 0.6 mL/min), the proportion of mobile phase acetonitrile/water (60:40 and 70:30 *v*/*v*), and the pH (3 or 4). The tested parameters and levels are shown in Table 5. Robustness was performed in triplicate from injections of sample solutions containing 25 µg/mL of RA standard.

#### 3.3.6. Accuracy

The accuracy of the method was assessed through recovery tests, by quantifying RA in freshly prepared solutions of standard RA and commercial RA. For this analysis, quantities of commercial RA were added to standard RA, obtaining concentrations of 7, 20, and 30 µg/mL. The recovery percentages were calculated using Equation (4):(4)$$\%R=\frac{Ca-Cna}{Ctp}\times 100$$
where *R* stands for recovery, *Ca* is the concentration of RA determined experimentally in each of the prepared solutions (obtained by mixing RA commercial with RA standard) (µg/mL), *Cna* is the concentration of RA determined experimentally in the standard RA solutions (µg/mL) without commercial RA, and *Ctp* is the theoretical concentration of RA (7, 20 and 30 µg/mL) added to the sample.

### 3.4. Ricinoleic Acid (RA) Content (Commercial Sample) by Gas Chromatography

To determine the RA content of the sample used in the development of nanocapsules (supplied from Azevedo Oléos, SP, Brazil), gas chromatography was used. Fatty acid methyl esters were separated and determined by a gas chromatograph (Shimadzu^®^, model GC 2010, Kyoto, Japan), equipped with a fused silica capillary column (CP—BPX-70, 30 m × 0.25 mm i.d. and 0.25 μm) and flame ionization detector (FID). The carrier gas used was helium. Injections were performed using an AOC-20i (automatic sampling system), equipped with a 10 µL syringe. The injected volume was 1 µL in Split 50:1 mode. The temperature of the injector and detector was set at 250 °C. The initial column temperature was established at 80 °C for 3 min; then, the temperature increased at a rate of 10 °C/min until reaching 140 °C. Following this, the heating ramp was increased at a rate of 5 °C/min, until reaching 250 °C, remaining in this range for 5 min, accounting for a total of 40 min of analysis. Retention times and areas were automatically computed in GC Solution in Shimadzu’s Lab solutions software (Shimadzu, Kyoto, Japan). For analysis via gas chromatography, samples were prepared using 1 g of the diluted standard (Cayman Chemical, Ann Arbor, MI, USA) in 50 mL of hexane, reaching a concentration of 0.02 g/mL (20,000 ppm), and 1 g of RA (Azevedo Óleos, SP, Brazil) diluted in 50 mL of hexane, reaching a concentration of 0.02 g/mL (20,000 ppm). Subsequently, a solution at a concentration of 50 ppm was made for analysis on the gas chromatograph.

### 3.5. Preparation of Ricinoleic Acid (RA)-Loaded Nanocapsules

RA-loaded nanocapsules with different concentrations (Table 5) and bioactivities (NCRA1, NCRA2; NCRA3 and NCRA4, nanocapsules of PLGA without RA (Blank NP)) were developed using the emulsion-solvent evaporation method [18]. The organic phase was prepared by dissolving PLGA, Span 60^®^, and RA in ethyl acetate. The aqueous phase was prepared with 1% (*m*/*v*) of PVA in distilled water. The aqueous phase was heated in a water bath at 60 °C for the complete solubilization of PVA. Then, the organic phase was dropped to the aqueous phase, and the mixture was homogenized for 10 min at 15,000 rpm (Ultra-Turrax T25 basic Ika, Works, Wilmington, NC, USA) to form an oil/water emulsion. The emulsion was placed in a vacuum rotary evaporator (Model 803, Fisatom, SP, Brazil) for 30 min to remove the organic solvent. Subsequently, the solution was centrifuged at 18k rpm, at 4 °C. Nanocapsules were separated from the liquid and washed three times with distilled water to remove non-solubilized polymers.

### 3.6. Determination of Particle Size, Polydispersity Index, and Zeta Potential

The nanoparticle size and size distribution (polydispersity index) were measured through dynamic light scattering (DLS) using a Zetasizer Nano Zs (Malvern Panalytical, Malvern, Worcestershire, UK). A 4 mW HeNe (633 nm) laser source was used, at an angle of 90° and temperature of 25 °C. For analysis, 10 µL aliquots were diluted in 1 mL of distilled water and placed in polystyrene disposable cuvettes. Three replicates were performed, and the mean ± standard deviation was reported. The zeta potential (ZP) was measured by electrophoretic light scattering (ELS), using a Zetasizer Nano ZS (Malvern Instrument, Worcestershire, UK). A polycarbonate disposable cell with copper electrodes was used. Samples were diluted 1:9 (*v*/*v*) in Milli-Q water, and measurements were performed at 25 °C and 150 V. Three replicates were performed, and the mean ± standard deviation was reported. The graphics were processed using the software Origin^®^ 8.5.

### 3.7. Encapsulation Efficiency

The encapsulation efficiency of RA-loaded nanocapsules was determined by centrifugation method [59]. For this, 500 µL of nanosuspension was filtered using a 10 kDa centrifugal filter (Amicon^®^ Ultra, Millipore, Billerica, MA, USA) and centrifuged for 30 min at 14.000 rpm in a refrigerated thermo centrifuge (Heraeus Megafuge 16R (Thermo Fisher Scientific Inc., Waltham, MA, USA)). After that, 250 µL of the supernatant was removed and added to 500 µL of mobile phase (acetonitrile: water, 65:35 *v*/*v*) to be analyzed by HPLC. The RA concentration was calculated using an appropriate analytical curve, considering the dilution factor. The encapsulation efficiency was performed in triplicate and calculated using Equation (5).
(5)$$EE\%=\frac{Total\;amount\;of\;drug\;\left(g\right)-Free\;drug\;(g)}{Total\;amount\;of\;drug\;\left(g\right)}\times 100$$

### 3.8. Nanoparticle Morphology Characterization

The morphology of the nanocapsules was evaluated by transmission electron microscopy (TEM). Five (5) μL of the nanosuspension was placed onto carbon-coated copper grids and left to dry in a silica desiccator for 16 h. The grid was then negatively stained using a 2% uranyl acetate solution. TEM images were taken using a JEOL JEM 2011 microscope (JEOL, Tokyo, Japan), operated at an accelerating voltage of 120 kV.

## 4. Conclusions

A new method utilizing HPLC-DAD was developed and validated for the identification and assessment of encapsulation efficiency in polymeric nanocapsules produced through the emulsion-evaporation technique. The nanocapsules exhibited ideal properties, such as appropriate particle size, PDI, and ZP, contributing to the system’s stability and achieving high encapsulation efficiency. The analytical method demonstrated simplicity and suitability for the detection and quantification of ricinoleic acid (RA), making it applicable in developing new formulations containing this active ingredient, which has promising antimicrobial and anti-inflammatory activities. This validated method for determining ricinoleic acid was successfully validated and can be used in the future to determine this bioactive in more complex matrices, enhancing its application in various fields.

## Figures and Tables

**Figure 1 pharmaceuticals-17-01220-f001:**
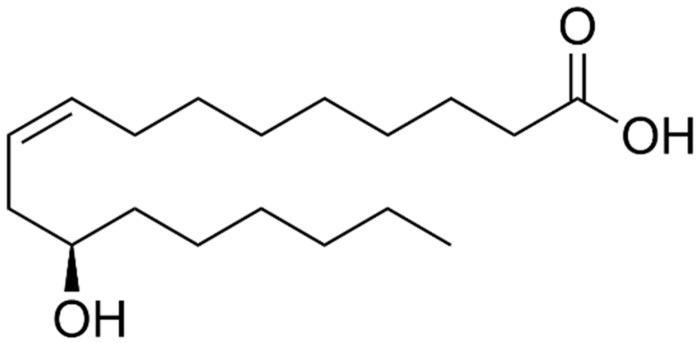
Chemical structure of ricinoleic acid (RA) (12-hydroxyoctadec-9-enoic acid).

**Figure 2 pharmaceuticals-17-01220-f002:**
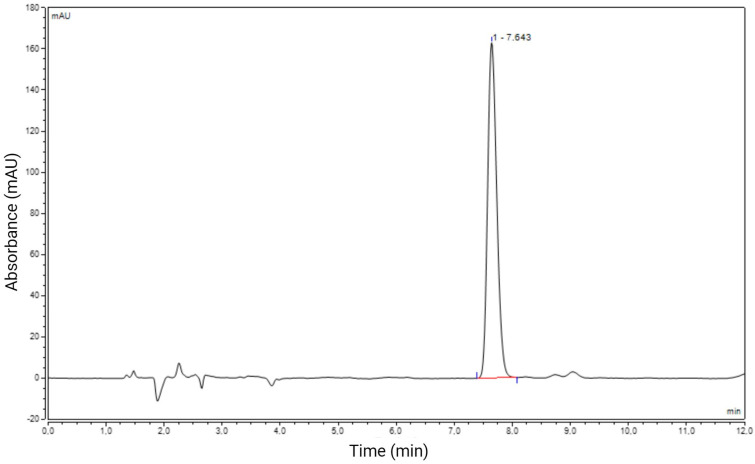
HPLC chromatogram of 50 μg/mL ricinoleic acid (RA) standard solution. Conditions: mobile phase acetonitrile: water (65:35, *v*/*v*); flow rate 0.8 mL/min; column temperature 25 °C and injections volume 20 µL.

**Figure 3 pharmaceuticals-17-01220-f003:**
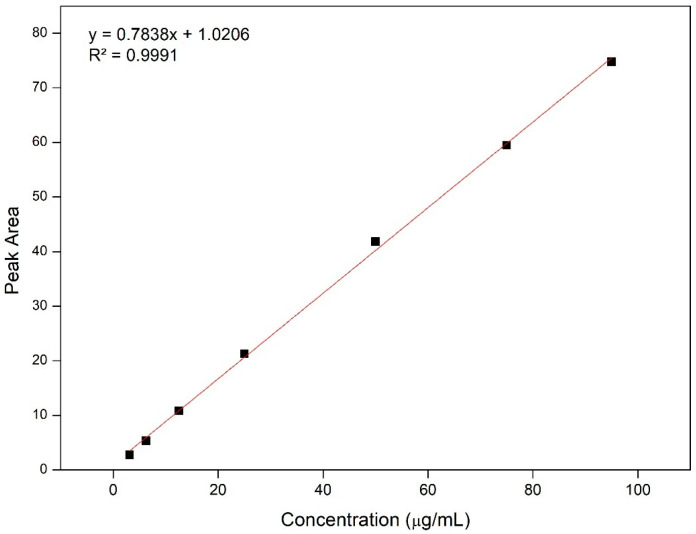
Standard calibration curve (y = 0.7838x + 1.0206) of ricinoleic acid.

**Figure 4 pharmaceuticals-17-01220-f004:**
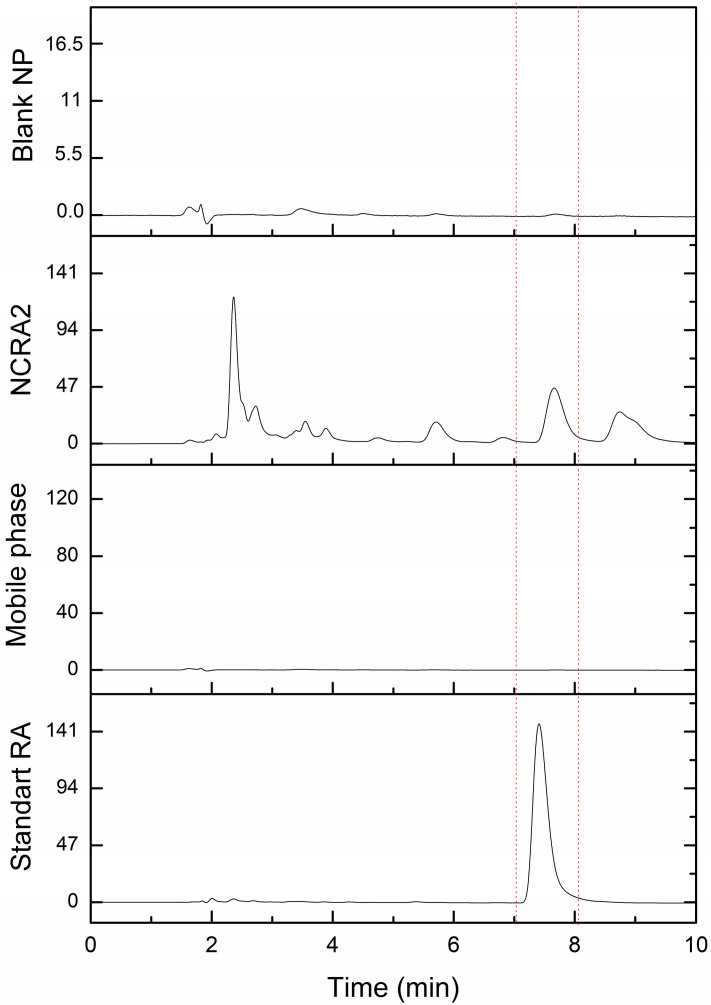
HPLC chromatogram of standard RA, mobile phase, blank NP (nanocapsules without RA) and sample NCRA2 (polymeric nanocapsules containing RA).

**Figure 5 pharmaceuticals-17-01220-f005:**
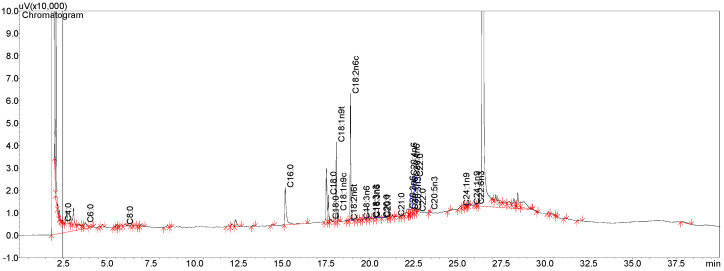
Chromatogram (GC) of ricinoleic acid (commercially obtained from Azevedo óleos).

**Figure 6 pharmaceuticals-17-01220-f006:**
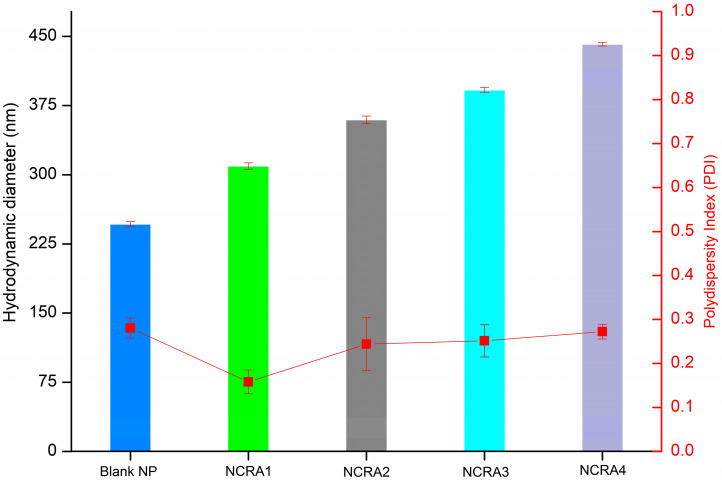
Hydrodynamic diameter and polydispersity index measurements of polymeric nanocapsules containing ricinoleic acid (RA). Size results for different nanocapsules with different concentrations of RA (Captions: Blank NP, nanoparticles without RA; NCRA1, nanocapsules containing 10 mg/mL of RA; NCRA2, nanocapsules containing 10 mg/mL of RA; NCRA3, nanocapsules containing 20 g/mL of RA, and NCRA4, nanocapsules containing 40 µg/mL of RA.

**Figure 7 pharmaceuticals-17-01220-f007:**
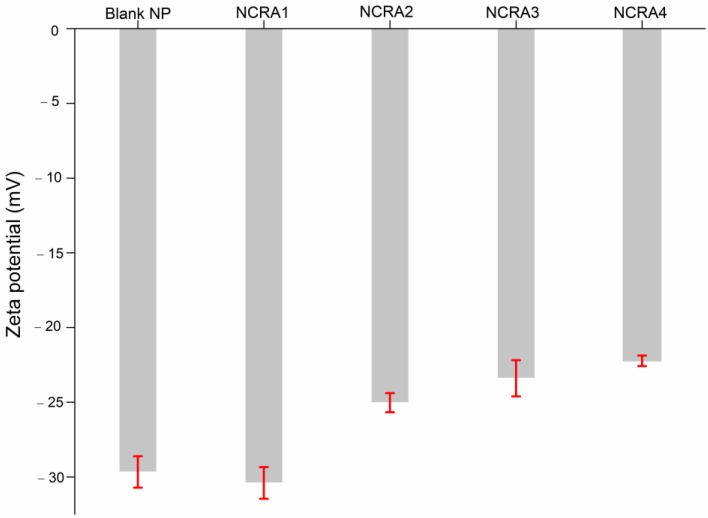
Zeta potential results of for different nanocapsules with different concentrations of ricinoleic acid (RA). The error bars represent standard error of three individual measurements. Captions: Blank NP, nanoparticles without RA; NCRA1, nanocapsules containing 10 mg/mL of RA; NCRA2, nanocapsules containing 10 mg/mL of RA; NCRA3, nanocapsules containing 20 g/mL of RA, and NCRA4, nanocapsules containing 40 µg/mL of RA.

**Figure 8 pharmaceuticals-17-01220-f008:**
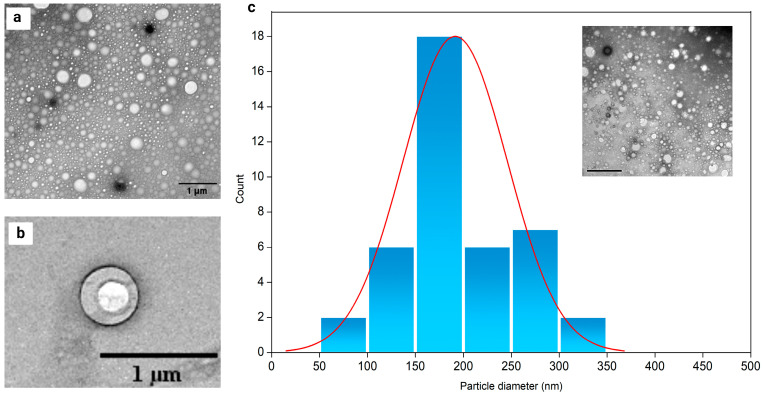
TEM images of negatively stained (**a**,**b**) and size distribution (**c**) obtained from TEM images by ImageJ 1.8.0 Software for PLGA nanocapsules containing RA (sample NCRA1).

**Table 1 pharmaceuticals-17-01220-t001:** Results of the validation parameter: precision.

Validation Parameters	Results
Intra-assay precision (Repeatability) (*n* = 6 determinations) Theoretical concentration (µg/mL) Concentration obtained (µg/mL)	1.93%
	25.00
	26.01 ± 0.48
Inter-assay precision (*n* = 12 determinations) Theoretical concentration (µg/mL) Concentration obtained (µg/mL)	2.43%
	25.00
	27.05 ± 0.65

RSD = relative standard deviation.

**Table 2 pharmaceuticals-17-01220-t002:** Results of the parameters evaluated to analyse the robustness of the analytical method.

Parameters	Variation	Concentration (µg/mL)	Area	RSD (%) Area	Ret. Time (min)	RSD (%) Ret. Time	Symetry Factor	RSD (%) Symettry Factor
Flow (mL/min)	1.0	23.55	19.745	1.379	6.02	1.053	1.68	0.907
	0.8 *	28.72	23.532	0.764	7.49	0.231	1.71	0.336
	0.6	38.34	31.702	1.005	10.04	0.304	1.73	0.879
Mobile phase ACN:H_2_O	60:40	32.09	26.177	1.4041	10.23	0.564	1.78	0.854
	0.8 *	31.27	23.532	0.764	7.49	0.231	1.71	0.336
	70:30	39.61	32.068	0.185	5.86	0.984	2.63	4.055
pH	4	34.89	28.3657	0.823	7.64	1.022	1.71	0.336
	3 *	28.72	23.532	0.764	7.49	0.231	1.71	0.336

* Normal conditions.

**Table 3 pharmaceuticals-17-01220-t003:** Accuracy results of ricinoleic acid (RA) by the HPLC method.

Ricinoleic Acid (RA) Concentration (µg/mL)	Recovery (%)	RSD (%)
7.90 ± 0.14	100.10 ± 2.56	3.14
20.35 ± 0.28	96.90 ± 1.74	
32.10 ± 0.469	103.70 ± 1.92	

RSD = relative standard deviation.

**Table 4 pharmaceuticals-17-01220-t004:** Encapsulation efficiency for PLGA nanocapsules (NPs) containing ricinoleic acid (RA).

Sample	Theorical Concentration (mg/mL)	Experimental Concentration (mg/mL)	EE%
NCRA1	40.00	39.92	99 ± 0.14
NCRA2	100.00	98.51	99 ± 0.49
NCRA3	200.00	199.92	99 ± 0.06
NCRA4	400.00	399.90	99 ± 0.39

**Table 5 pharmaceuticals-17-01220-t005:** Composition of the developed polymeric nanocapsules.

Samples	Aqueous Phase	Organic Phase			
	PVA 1% (mL)	PLGA (mg)	Span 60^®^ (mg)	Ethyl Acetate (mL)	Ricinoleic Acid (mg)
Blank NP	10	50	10	2	-
NCRA1	10	40	10	2	100
NCRA2	10	50	10	2	100
NCRA3	10	50	10	2	200
NCRA4	10	50	10	2	400

## Data Availability

Data is contained within the article or Appendix A.

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
