# Peer review of "A Newly Validated HPLC-DAD Method for the Determination of Ricinoleic Acid (RA) in PLGA Nanocapsules"

_pharmaceuticals, 2024, doi:10.3390/ph17091220_

Round 1

Reviewer 1 Report

Comments and Suggestions for Authors

Thank you for the opportunity to review the manuscript. 

The manuscript has a clear structure.

In my opinion, the manuscript has two main issues that the authors need to address in order to improve its quality and significance:

1. The authors need to emphasise the advantages of the HPLC method they have developed and validated over the already published work in this submission [Characterisation of Castor Oil by HPLC and Charged Aerosol Detection. Marc Plante, Chris Crafts, Bruce Bailey and Ian Acworth].

2. The authors prepared and characterised a nanoformulation, but did not give the reason why the nanoparticles were formulated.

Here I see several ways to strengthen the practical side of the manuscript:

a) The manuscript needs to be supplemented with a set of biological experimental results according to the proposed application. For example, if the nanoformulation is proposed for GI delivery, appropriately designed release, ricinoleic acid (RA) stability (after loading into NPs), in vitro toxicity test results, in vitro ROS levels (general levels) are needed. The nanoparticles are already partially characterised.

b) The authors should perform the comparison of particles of different nature (e.g. albumin and chitosan or gelatine and alginate) loaded with RA, not only PLGA, using the HPLC developed and evaluated by them (and including the extraction method).

c) The HPLC method was developed and validated to analyse RA content and stability in different samples - so, it would be informative to prove this method on several (2-4) samples of castor oil formulations available on the market (in this case the samples can be anonymized). 

Comments on the Quality of English Language

The English is generally good, but I met rare mistakes - the manuscript should be proofread carefully.

Author Response

Dear reviewer, thank you very much for your suggestions for the development of this paper. Please check the attachment.

Reviewer 2 Report

Comments and Suggestions for Authors

Dear Editor

I am writing to provide my review of the manuscript titled "A Newly Validated HPLC-DAD Method for the Determination of Ricinoleic Acid in PLGA Nanocapsules".  The research focuses on developing and validating a high-performance liquid chromatography with diode-array detection (HPLC-DAD) method for identifying and quantifying ricinoleic acid in polymeric nanocapsules.

The paper primarily appears to follow routine procedures without introducing significant novelty. The authors should clearly articulate the innovative aspects of their work.

1.   The abstract doesn't mention the scope clearly at the beginning. It's beneficial to state the specific aim right away.

2.   No need for this sentence in the abstract (However, its pharmacological effects can be affected by factors such as high temperatures, humidity, and light exposure.)

3.   Line 36, (oil encapsulation ) specify the oil name

4.   Line 31, the author mention 1H-NMR, but no any details for this study

5.   Your introduction provides a comprehensive overview of castor oil and its constituents, particularly ricinoleic acid. However, since the main focus of your study is on the optimization and validation of an HPLC-DAD method for ricinoleic acid detection and quantification, it would be beneficial to reduce the background information on castor oil and concentrate more on ricinoleic acid and the HPLC method.

6.   The introduction lacks a detailed explanation of the necessity and novelty of the proposed HPLC-DAD method. The authors should elaborate on how their method advances the current state of analytical techniques for ricinoleic acid determination.

7.   Compare your method with existing methods in the literature to highlight any unique features or improvements.

8.   Ensure terminology is consistently used throughout the text. For instance, using "ricinoleic acid (RA)" consistently after its first mention.

9.   Line 58, (cultivated in Brazil the 2018/2019 and harvest, mainly, ) correct the sentence .

10.  LD50, should be LD50

11.  The term "ricinoleic acid" is repeated multiple times. Use the abbreviation (RA) after the first mention to avoid redundancy

12.  At the end of the introduction the author should state the aim and the novelty of the study .

13.  Line 124-126, (method was validated for the accurate determination of ricinoleic acid encapsulated within PLGA nanocapsules equipped 125 with a diode array detector (DAD) model Ultimate 3000 RS Variable Wavelength,) not clear, rewrite it.

14.  Line 132, (Detection  was performed at 200, 205, 210, and 215 nm) at what point exactally the detection done .?

15.  Line 131, (acetonitrile: water (65:35 v/v)  based on what the author select this solvent mixture and why not methanol: water is used .

16.  Linearity, the standard curve was constructed by preparation solution of ricinoleic acid in acetonitrile (50:50 v/v). why not same as the ratio used in the method development .

17.  Line 151, (The encapsulation efficiency of the process was determined calculating the difference between the free ricinoleic acid and the total amount of it used for preparation.) the senteance not clear, revise it.

18.  Line 256, (acetonitrile:water, 50:50 v/v) why no (65:35 v/v) ?.

19.  The figure 2, is not clear and very small.

20.  Line 328, (To evaluate the specificity was evaluated by comparing) revise it , not clear.

21.  The code NCRA2  and ( NRA 7.2) are not consistent ,

22.  Line 330, (To extraction of RA of nanoparticles (NRA 7.2),) the sentence not clear, revise it.

23.  Also the figure 4, poor resolution, improve it

24.  What was the concentration used for study Robustness and other parameters?

25.  Line 391, (DPR of 3.14%. ) DPR abbreviation for what ?

26.  Line 452, (Studies carried out by Saxena et al., 2004, Abd El Hady et al.,  2023, and Amin & Boateng, 2023, obtained similar values of zeta power (ζ) for PLGA) but there is no reference ?

27.  Do you have figure for size and zeta of the prepared nanoparticles ?

28.  Line 472, (a study carried out by Froiio et al., 2019, who) where the reference ?

29.  There are typos and grammatical errors throughout the manuscript, a thorough proofread is needed for correction.

Comments on the Quality of English Language

There are typos and grammatical errors throughout the manuscript, a thorough proofread is needed for correction.

Author Response

(The authors gave the same response as above.)

Round 2

Reviewer 2 Report

Comments and Suggestions for Authors

I have some concerns regarding the responses provided by the authors to my comments.

1.       Your introduction provides a comprehensive overview of castor oil and its constituents, particularly ricinoleic acid. However, since the main focus of your study is on the optimization and validation of an HPLC-DAD method for ricinoleic acid detection and quantification, it would be beneficial to reduce the background information on castor oil and concentrate more on ricinoleic acid and the HPLC method.

There was no response from the authors addressing this point.

2.       The introduction lacks a detailed explanation of the necessity and novelty of the proposed HPLC-DAD method. The authors should elaborate on how their method advances the current state of analytical techniques for ricinoleic acid determination.

The author response for this comment was by addition (line 105-121 ) this data not needed and should be deleted, the author should understand what I mean by the comment it general word about HPLC delete it , (HPLC method, I mean the method that have done for  ricinoleic acid for detection and qualification, the other method, and why your method is better or the mention your novelty of your HPLC method

3.       Compare your method with existing methods in the literature to highlight any unique features or improvements.  

No response for this comment from the author. Why ?

4.       Ensure terminology is consistently used throughout the text. For instance, using "ricinoleic acid (RA)" consistently after its first mention.

I pointed out the need for consistent terminology, such as using "ricinoleic acid (RA)" consistently after its first mention. The authors have not addressed this issue, and the manuscript still contains inconsistencies.

5.       At the end of the introduction the author should state the aim and the novelty of the study

No response from the author  

6.       Line 132, (Detection  was performed at 200, 205, 210, and 215 nm) at what point exact ally the detection done .?

The author responded by selecting 240 nm. Can the author provide approve (figure for the UV wavelength peak )  for the maximum wavelength for the absorption ?

7.       Do you have figure for size and zeta of the prepared nanoparticles ?

I requested a figure showing the size and zeta potential of the prepared nanoparticles from the Zetasizer instrument. The authors provided a curve created from Excel, which is not an appropriate response. A peak from the Zetasizer instrument should be provided.

8.       There are typos and grammatical errors throughout the manuscript, a thorough proofread is needed for correction.

Round 3

Reviewer 2 Report

Comments and Suggestions for Authors

The authors now improve the article and response to all my concern